# Anomalies are Streaming: Continual Learning for Weakly Supervised Video Anomaly Detection

## Abstract

Weakly supervised video anomaly detection (WSVAD) aims to locate frame-level anomalies with only video-level annotations provided. However, existing WSVAD methods struggle to adapt to real-world scenarios, where unseen anomalies are continuously introduced, thereby making the training of WSVAD essentially a process of continual learning. In this paper, we pioneer to explore the continual learning for weakly supervised video anomaly detection (CL-WSVAD), seeking to mitigate the catastrophic forgetting when the detection model learns new anomalies. We propose normality representation pre-training prior to continual learning, utilizing potential anomaly texts to guide the model in learning robust normality representations, which improves discrimination from potential incremental anomalies. Additionally, we introduce a mixed-up cross-modal alignment method to assist in adapting the pretrained model on CL-WSVAD. Subsequently, we propose a continual learning framework based on sequentially retaining the learnable text prompts for each type of anomaly, which effectively mitigates catastrophic forgetting. Experiments on our established CL-WSVAD benchmarks demonstrate the superiority of proposed method.

## 1 Introduction

Video anomaly detection (VAD), which aims to identify uncommon events or behaviors in video sequences that deviate from usual patterns, is widely applied in public security, intelligent surveillance, and evidence investigation (Benezeth et al., 2009). VAD methods are typically designed to automatically predict frame-level anomaly scores over the timeline of the video. Due to the rarity of abnormal samples and the expensive frame-level annotations, mainstream paradigms are unsupervised video anomaly detection (UVAD) (Liu et al., 2018; Lee et al., 2019; Lv et al., 2021; Zhang et al., 2024) and weakly supervised video anomaly detection (WSVAD) (Sultani et al., 2018; Wu et al., 2020; Tian et al., 2021; Wu et al., 2024b).

UVAD methods learn normal patterns from only normal data and identify the videos deviating from this distribution as anomalies. Due to the exclusion of anomalies during training, the UVAD methods exhibit insufficient generalization performance in complex scenarios (Yang et al., 2024). Subsequently, WSVAD introduces anomalous videos and video-level labels to guide the model to learn the discrimination between normal and abnormal instances. WSVAD leverages Multiple Instance Learning (MIL) by constructing positive bags, each containing at least one anomalous frame, to train the model to infer which specific segments are anomalous within positive bags, without relying on frame-level annotations (Sultani et al., 2018; Wu et al., 2020).

While the performance of WSVAD has been continuously improved, an inherent flaw of WSVAD has been consistently ignored. In real-world VAD scenarios, not all anomalies are necessarily provided for training all at once. Conversely, anomalous data indeed are continuously supplemented as training data for updating the model. Apparently, the WSVAD model, which is trained once on the limited dataset, lacks the ability for continual learning (CL). Consequently, WSVAD methods tend to overfit to known anomalies and exhibit limited generalization to unknown anomalies. Therefore, WSVAD training in real-world scenarios is more appropriately a anomaly-incremental continual learning process. However, when an unseen abnormal category is captured, continuously mixing

this upcoming abnormal data with the original training set for retraining is an inefficient training approach. More importantly, due to the privacy concerns associated with anomalies captured in specific scenarios, previously acquired anomalies are not necessarily accessible for data security issues. However, solely applying the new abnormal data to fine-tune the model results in the original knowledge of the model being overwritten, leading to catastrophic forgetting (McCloskey & Cohen, 1989). Fortunately, continual learning (Li & Hoiem, 2017; Wang et al., 2023) presents a potential solution to this issue. Doshi & Yilmaz (2020; 2022) address CL for UVAD and achieve the expected performance, but CL for WSVAD with anomaly-incremental process remains unexplored.

Nevertheless, directly applying existing CL methods to WSVAD raises two issues. Firstly, CL methods are typically applied to class-unbiased incremental tasks (Thengane et al., 2022; Wang et al., 2023). However, normal instances, which are basically used as the golden standard to define the anomalies, require special considerations in learning their representations especially in the continual setting, where the anomalies are streaming and diverse in categories. Therefore, learning a robust normality representation is crucial for the continual learning in WSVAD. Secondly, the classical CL methods, relying on either data replay or parameter isolation, suffer from increasing memory units (Isele & Cosgun, 2018; Rolnick et al., 2019; Shin et al., 2017) or model size (Aljundi et al., 2017; Mallya & Lazebnik, 2018; Serra et al., 2018).

In this work, we pioneer to explore continual learning for WSVAD, aiming to address the aforementioned two issues. In the CL-WSVAD paradigm, normal videos and one type of anomalous video are provided in the initialization task, and each subsequent task sequentially introduces a new anomalous type. **For the initialization task**, we further decompose it into stages of normality representation pre-training and weakly supervised adaption. On the normality representation pre-training stage, we leverage the strong vision-language alignment in CLIP with readily available yet rich texts describing anomalies as a complementary for anomalous videos in model pre-training, preparing the enhanced normality representations for the adaption stage. To facilitate the model to extract meaningful representations for both normalities and abnormalities, we in the weakly supervised adaption stage propose the mixed-up cross-modal alignment method, which aligns visual features and textual embeddings on the normality-abnormality mixed image-text pairs. **In the anomaly continual learning stage**, we design a novel continual learning framework by introducing a set of learnable text prompts while fixing the other model parameters to mitigate catastrophic forgetting. Particularly, we maintain these text prompts exclusively for each subsequent task, avoiding the large-scale memory and model expansion. Compared to UVAD, CL-WSVAD introduces anomalous data to handle complex scenarios. In contrast to WSVAD trained on fixed datasets, CL-WSVAD enhances the scalability of WSVAD to adapt to continuously introduced anomalies, addressing the challenge of exhaustively collecting anomalies. Additionally, CL-WSVAD, as an improved paradigm based on WSVAD, incurs no additional costs for data collection and annotations compared to WSVAD, yet it learns new anomalies without relying on previous ones, ensuring data privacy. Furthermore, our proposed method offers better efficiency as it only requires updating prompts with minimal parameters on newly introduced data.

Our contributions are summarized as follows:

- We pioneer to explore the method for addressing with streaming anomalies in the real world, proposing the new paradigm: continual learning for weakly supervised video anomaly detection (CL-WSVAD).

- We specifically propose a normality representation pre-training method for CL-WSVAD, which guides the detection model to first learn a general normality representation to enhance the discrimination between normal and potential incremental anomalies. Additionally, a mixed-up cross-modal alignment method is proposed to guide the pre-trained model in achieving effective adaptation on CL-WSVAD.

- We design a novel CLIP-based continual learning framework, which sequentially maintains the learnable text prompt corresponding to each task, mitigating the catastrophic forgetting in CL-WSVAD.

- We compared our method with existing continual learning methods and achieve superior performance on mainstream datasets. Extensive experiments validate the effectiveness of our method in continual learning.

## 2 RELATED WORK

**Weakly Supervised Video Anomaly Detection.** In the weakly supervised video anomaly detection paradigm, a pre-trained video backbone is utilized to extract features from video segments, followed by training a temporal anomaly detector to predict anomaly scores for the video segments. Sultani et al. (2018) introduce a large-scale real-world surveillance video dataset, UCF-Crime, and propose a MIL based ranking loss to enhance the discrimination between abnormal segments and normal segments. Wu et al. (2020) introduce graph convolutional networks to extract the dependencies between video segments in both feature context and temporal distance, and fuse video-audio information to enhance the performance of anomaly detection. RTFM (Tian et al., 2021) and MGFN (Chen et al., 2023) explore the correlation between feature magnitude and abnormal segments, leveraging this correlation to enhance the discrimination between abnormal and normal features. With vision-language models achieving superior results in visual tasks, VadCLIP (Wu et al., 2024b) transfers the pre-trained CLIP to WSVAD, where pre-trained language-visual knowledge effectively enhances detection performance. Yang et al. (2024) transfer the language-visual knowledge of CLIP model for aligning the video text descriptions and corresponding video frames to generate more accurate pseudo labels, which guide the model to achieve better self-supervised model training. Tao et al. (2024) propose a novel multi-prompt learning strategy, where the textual abnormal event prompts extracted from generated video descriptions are utilized to implicitly guide the model in learning the definition of anomalies. Lv & Sun (2024) adapt Video-LLaMA to the WSVAD task, achieving not only threshold-free anomaly detection but also providing explanations for anomaly alerts. Jain et al. (2025)presents a practical cross-domain learning framework for WSVAD and employs unlabeled external videos to enhance the cross-domain generalization of the model. However, existing WSVAD methods are typically based on the once training setting, failing to address the fact that real-world anomalies are streamly introduced for model updates. In this paper, we pioneer to explore the continual learning for WSVAD, aiming to mitigate the catastrophic forgetting when continuously introducing previously unseen anomalies.

**Continual Learning.** Continual learning, which is a learning paradigm designed for an infinite stream of data, strives to incrementally expand acquired knowledge for future learning (De Lange et al., 2021). The existing continual learning methods can be mainly categorized into three categories: replay methods (Isele & Cosgun, 2018; Rolnick et al., 2019; Buzzega et al., 2020), parameter isolation methods (Serra et al., 2018; Xu & Zhu, 2018), and regularization-based methods (Aljundi et al., 2018; Li & Hoiem, 2017; Dhar et al., 2019) . Replay methods employ stored samples when learning new tasks to mitigate catastrophic forgetting. Parameter isolation methods design separate sub-models for each task to mitigate catastrophic forgetting of previous tasks. However, the continuously increasing stored samples and the expanding model size severely limit the extensibility for continual learning, making them evidently unsuitable for WSVAD. Regularization-based methods add explicit regularization terms on weights or data to guide the model in consolidating previous knowledge while learning new tasks. Nonetheless, these regularization-based approaches constrain the performance of the model on new tasks. Recently, CLIP based methods, such as Continual-CLIP (Thengane et al., 2022) and AttriCLIP (Thengane et al., 2022; Wang et al., 2023), achieve promising results in CL without sample storage and extensive model expansion. Unfortunately, anomalies in WSVAD are complex and diverse, these methods challenge in adapting to WSVAD and achieving expected performance. Based on the characteristics of VAD, we propose a continual learning method that emphasizes on learning general normality representation, achieved by differentiating normal videos from abnormal texts. To facilitate CLIP adaption to WSVAD which involves capturing various degrees of anomalies, we propose a cross-modal alignment based on mixed-up anomalies with various mix-up factors. Unlike AttriCLIP which updates prompts throughout the training, we develop a novel continual prompt learning framework, which sequentially retains the learnable text prompts for each task, effectively mitigating catastrophic forgetting.

## 3 APPROACH

### 3.1 PRELIMINARIES

In the WSVAD paradigm, untrimmed training videos $\{v_n\}_{n=1}^N$ and corresponding video-level labels $\{y_n\}_{n=1}^N$ are provided in the training stage. Here, the video which entirely lacks abnormal frames is labeled as normal video with $y_n = 0$, while the video containing at least one abnormal frame

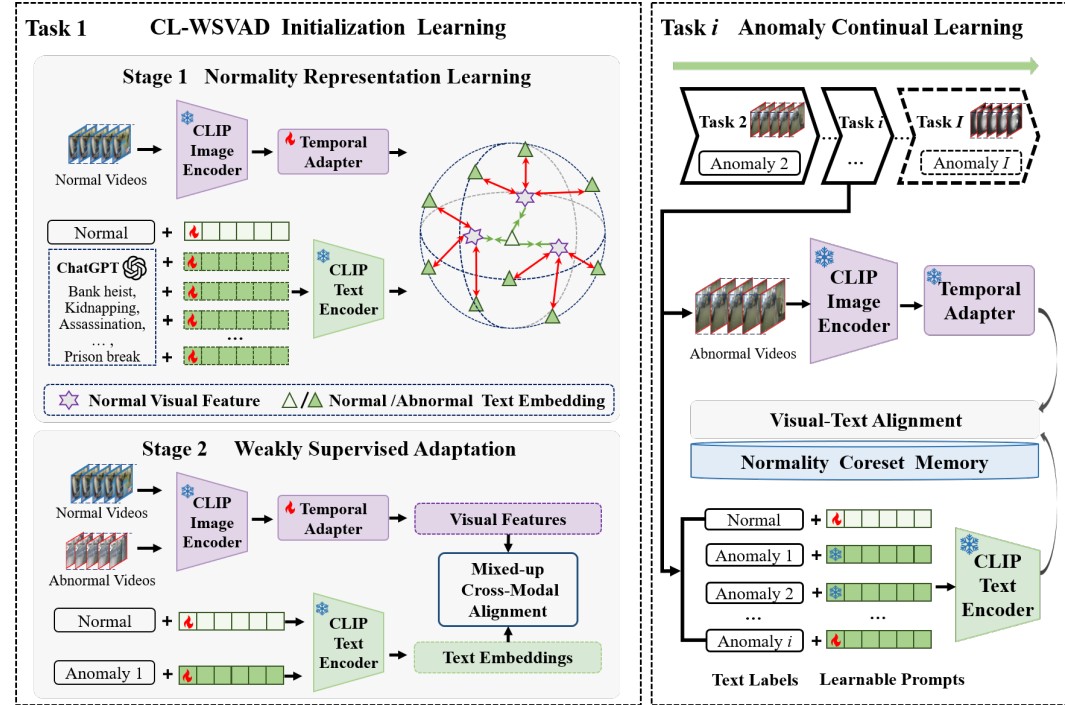

Figure 1: Overall framework of our method. In Task 1, normality representation pre-training guides the detection model to learn a robust normality representation. Then mixed-up cross-modal alignment assists the pre-trained model in adapting to CL-WSVAD. In Task $i$ ($i > 1$), anomalies are streamingly introduced, and the learnable text prompts corresponding to each task are trained and retained sequentially to mitigate catastrophic forgetting. Note that the modules marked with snowflakes are frozen, while those marked with flames are trained. In Task $i$, the parts with solid lines indicate previous and ongoing tasks, while the parts with dashed lines represent subsequent tasks.

is labeled as abnormal video with $y_n = 1$. Generally, video $v_n$ is firstly divided into $T_n$ non-overlapping segments, *i.e.*, $v_n = \{v_{n,t}\}_{t=1}^{T_n}$, and each video segment $v_{n,t}$ is fed into the pretrained feature extractor to extract video features. Then, a temporal anomaly detector is weakly supervised trained to predict frame-level anomaly scores.

## 3.2 CL-WSVAD FORMULATION

Since anomalies in the real world emerge continuously, WSVAD is more inclined towards an abnormal-class-incremental learning task. In this paper, we pioneer to propose the new paradigm: continual learning for weakly supervised video anomaly detection. Specifically, given a sequence of tasks, Task={Task 1, Task 2, ..., Task $I$}, with corresponding datasets $\mathcal{D}=\{\mathcal{D}_1, \mathcal{D}_2, ..., \mathcal{D}_I\}$, the datasets are sequently and non-overlappingly fed into the continuous tasks. Due to privacy and security concerns, in the $i^{th}$ task, the anomalies in previous dataset, $\{\mathcal{D}_1, \mathcal{D}_2, ..., \mathcal{D}_{i-1}\}$, are unavailable. Meanwhile, following the WSVAD paradigm, Task 1 provides normal videos and one type of anomaly video. In subsequent continuous tasks, each task introduces one type of anomaly video. The goal of continual learning for WSVAD is to mitigate the forgetting of knowledge from $\{\mathcal{D}_1, \mathcal{D}_2, ..., \mathcal{D}_{i-1}\}$ while learning new anomaly on $\{\mathcal{D}_i\}$.

## 3.3 CONTINUAL LEARNING FRAMEWORK

CLIP has been proven to be an efficient continual learner across multiple visual tasks (Thengane et al., 2022), and we adapt CLIP to CL-WSVAD, constructing a continual learning framework. CLIP consists of an image encoder $f_\theta$ and a text encoder $g_\phi$, and these two encoders respectively output image embedding $z$ and text embedding $w$. In the training stage, a contrastive loss is applied to align image embeddings with text embeddings. The prediction probability for the $i^{th}$ class can be

expressed as follows:

$$p_i = \frac{\exp(sim(\boldsymbol{z}, \boldsymbol{w_i})/\tau)}{\sum_{i=1}^{I} \exp(sim(\boldsymbol{z}, \boldsymbol{w_i})/\tau)}, \tag{1}$$

where $sim(\cdot, \cdot)$ denotes cosine similarity and $\tau$ is a temperature hyper-parameter for scaling.

To adapt CLIP to CL-WSVAD task, a GCN (Graph Convolutional Network) based temporal adapter, $f_a$, is introduced after the image encoder, constructing visual branch of the anomaly detection model. In this branch, the videos are non-overlappingly segmented and fed into the image encoder and the temporal adapter sequentially. In CL-WSVAD task, the prediction probability can be expressed as follows:

$$P_i = \frac{\exp(sim(f_a(Z), \boldsymbol{w_i})/\tau)}{\sum_{i=1}^{I} \exp(sim(f_a(Z), \boldsymbol{w_i})/\tau)}, \tag{2}$$

where $Z = \{z_1, ..., z_{T_n}\}$ represents the set of segment-level visual embeddings, and $P_i = \{p_i^1, p_i^2, ..., p_i^{T_n}\}$ denotes the set of segment-level predictions for the $i^{th}$ class.

Inspired by CoOp (Zhou et al., 2022), we introduce the adaptation strategy that fine-tunes the learnable text prompts, designing our continual learning framework. In Task $i$, the learnable prompt integrated input of text encoder is expressed as follows:

$$t_p^i = \{V_1^i, ..., V_M^i, Tokenizer(Labeli), V_{M+1}^i, ..., V_{2M}^i\}, \tag{3}$$

where $\{V_1^i, V_2^i, ..., V_{2M}^i\}$ are the learnable prompt containing $2M$ context tokens, and the $Tokenizer$ is the CLIP tokenizer. In initialization task, $f_a$ learns the dependencies among video segments and has acquired the ability to distinguish between normal and anomalous videos. Based on this observation, in subsequent continuous tasks, $f_a$ is frozen, and an independent learnable text prompt is provided for each task for vision-language alignment training. In Task $i$ ($i > 1$), the text learnable prompt $t_p^i$, which has been adapted by vision-language alignment, has already learned the current anomaly on Task $i$. Then, $t_p^i$ is frozen and is not trained in subsequent tasks. In this stage, a textual semantic contrastive loss $\mathcal{L}_{tsc}$ is introduced to enhance the discrimination between normal and anomalous text embeddings. $\mathcal{L}_{tsc}$ is represented as follows:

$$\mathcal{L}_{tsc} = \sum_i max(0, \frac{(\boldsymbol{w^n})^T \boldsymbol{w_i^a}}{\|\boldsymbol{w^n}\|_2 \cdot \|\boldsymbol{w_i^a}\|_2}), \tag{4}$$

where $\boldsymbol{w^n}$ and $\boldsymbol{w_i^a}$ represent the text embeddings for normal and $i^{th}$ anomaly, respectively. In this training strategy, each task retains independent $t_p^i$, and the text learnable prompts are independently trained without influencing other prompts. Therefore, this continual learning framework retains the streaming anomaly information and mitigates catastrophic forgetting. In the testing stage, the anomaly scores for video segments can be obtained by calculating the similarity between the normal / anomalous text embeddings and the visual features.

### 3.4 INITIALIZATION LEARNING

In our approach, Task 1 is designed as the initialization learning process of CL-WSVAD, which comprises two stages: the normality representation pre-training and the weakly supervised adaptation.

**Normality Representation Pre-training.** Different from traditional continual learning, we introduce normality representation pre-training (NRP) to obtain a robust normality representation. Although anomalous videos are scarce in the real world, fortunately, there is the perfect semantic alignment of vision and text features in CLIP. This semantic alignment allows easily accessible anomalous text to simulate real visual anomalies to guide the detector to distinguish between normal instances and potential incremental anomalies. Here, ChatGPT is utilized to generate potential anomalous texts, these generated texts are expected to cover a variety of potential anomalies. Specifically, ChatGPT is prompted with, "Please list possible abnormal events that may occur in videos", resulting in 2,000 potential abnormal texts. Then, a set of learnable text prompts are initialized with these anomalous texts to obtain the anomalous text embeddings. Then, a general normality representation is learned by contrastive learning on the actual normal visual features and potential anomalous text embeddings. The pre-training loss function $\mathcal{L}_{nrp}$ can be expressed as :

$$\mathcal{L}_{nrp} = \mathcal{L}_{nce} + \alpha \mathcal{L}_{tsc} = CE(y_{nor}, p_i^v) + \alpha \mathcal{L}_{tsc}, \tag{5}$$

where $\mathcal{L}_{nce}$ represents the cross-entropy loss between the predictions and the ground truth. $y_{nor}$ is the ground truth for normal videos, and $\alpha$ is a hyperparameter. Note that we first get segment-level predictions based on Eq. 2, and then employ the Top-K mean operation to obtain video-level predictions $p_i^v$. NRP guides the model to learn a generalized representation from extensive simulated visual anomalies, effectively distinguishing between normal and incremental anomalous events. In the Appendix A.1, we provide a theoretical proof of the effectiveness of NRP.

**Weakly Supervised Adaptation.** With a type of anomaly video is introduced, we adapt the pre-trained model to the CL-WSVAD paradigm. To enable the model to learning meaningful representations for both normalities and anomalies, we introduce the mixed-up cross-modal alignment method to assist in adaptation to CL-WSVAD.

Inspired by Wang et al. (2021); Mushtaq et al. (2024), the mixed features can incorporate the semantics of both components, based on which, mixed-up cross-modal alignment (MCMA) is proposed. Specifically, the normal embeddings, $\boldsymbol{z^n}$ and $\boldsymbol{w^n}$, and the anomalous embeddings, $\boldsymbol{z^a}$ and $\boldsymbol{w^a}$, which are respectively produced by the CLIP image encoder and CLIP text encoder, are mixed in the same proportion:

$$\boldsymbol{z^m} = \beta \boldsymbol{z^a} + (1-\beta)\boldsymbol{z^n}, \quad \boldsymbol{w^m} = \beta \boldsymbol{w^a} + (1-\beta)\boldsymbol{w^n}. \tag{6}$$

Here, $\boldsymbol{z^m}$ and $\boldsymbol{w^m}$ respectively represent the mixed visual and mixed text embeddings, and $\beta$ is the random mixing ratio factor. Then, the mixed visual embeddings are fed into the temporal adapter $f_a$, and the obtained visual features are expected to remain aligned with the mixed text embeddings in terms of anomaly semantics. To guide the temporal adapter in learning the anomaly semantics of the mixed visual features, a cross-modal alignment loss $\mathcal{L}_{cma}$ is introduced to guide the temporal adapter training. $\mathcal{L}_{cma}$ is expressed as follows:

$$\mathcal{L}_{cma} = m - (sim(\frac{1}{k}\sum\nolimits_{i=1}^{k} topk(f_a(\boldsymbol{z^m})), \boldsymbol{w^m})), \tag{7}$$

where $m$ is a constant representing the margin, and Top-K mean operation transforms segment-level mixup visual features into video-level features. By constructing numerous samples with varying degrees of anomalies using the mix-up technique, MCMA guides $f_a$ and the corresponding learnable text prompts to extract more meaningful normal and anomaly semantic information from the mixed features. Meanwhile, MCMA enhances the model's ability to effectively differentiate anomalies with varying levels of abnormality. In addition, this mix-up-based approach effectively augments the text and visual embeddings utilized for training, particularly enhancing the generalization of both normal and anomaly representations.

For the weakly supervised adaptation stage, with the introduction of a real abnormal type, the NRP is still applied to fine-tune the normality representation. The loss function $\mathcal{L}_{wsa}$ for this stage can be expressed as follows:

$$\mathcal{L}_{wsa} = \lambda \mathcal{L}_{cma} + \mathcal{L}_{nce} + \alpha \mathcal{L}_{tsc}, \tag{8}$$

where $\lambda$ and $\alpha$ are hyperparameters.

## 3.5 ANOMALY CONTINUAL LEARNING

In each subsequent task, a previously unseen category of anomalous videos is introduced for training. The parameters of the visual branch are frozen, and a dedicated learnable text prompt for current task is initialized and trained. Based on Eq. 2 and Top-K mean operation, video-level predictions can be obtained, and $\mathcal{L}_{nce}$ is utilized for optimization of the learnable text prompt.

**Normality Coreset Memory.** As we known, normal videos are widely available and easily accessible, without concerns regarding safety and privacy. To further improve the performance of proposed method, some representative normal features are saved as memory in Task 1 for subsequent tasks. Considering the substantial memory consumption of video features and the training efficiency, we propose the normality coreset memory (NCM). Specifically, the output features from the temporal adapter are first obtained. By comparing their cosine similarity with normal text embedding, the Top-K representative normal segment-level features are selected, and these features are then down-sampled to video-level normal features by mean operation. Subsequently, Greedy Coreset Subsampling (Roth et al., 2022) is employed to select coreset of normal video-level features, constructing the NCM in Task 1. With the introduction of NCM, the learnable text prompt corresponding to

Table 1: CL-WSVAD benchmark on UCF-Crime in AUC (%). The AUC of Task $i \in \{1, 2, ..., 13\}$, reports the AUC tested over all the previous tasks and the ongoing tasks (*i.e.*, Tasks 1, 2, . . . , $i$). $*$ denotes the reimplemented replay-based continual learning methods.

| Method | Task1 | Task2 | Task3 | Task4 | Task5 | Task6 | Task7 | Task8 | Task9 | Task10 | Task11 | Task12 | Task13 | Average |
|---|---|---|---|---|---|---|---|---|---|---|---|---|---|---|
| LWF | 99.8 | 95.4 | 95.0 | 89.4 | 87.6 | 82.7 | 82.7 | 80.3 | 79.3 | 78.8 | 74.4 | 75.1 | 73.9 | 84.2 |
| DER$^*$ | 99.7 | 93.0 | 93.7 | 89.6 | 87.4 | 82.6 | 82.5 | 81.8 | 81.7 | 79.9 | 79.2 | 80.0 | 80.1 | 85.5 |
| DER++$^*$ | 99.7 | 93.0 | 93.6 | 89.7 | 87.5 | 82.9 | 82.6 | 82.7 | 83.3 | 80.5 | 79.1 | 81.4 | 81.1 | 85.9 |
| Continual-CLIP | 99.8 | 97.3 | 96.5 | 93.1 | 90.6 | 86.3 | 86.1 | 85.3 | 84.6 | 83.0 | 81.1 | 81.4 | 81.4 | 88.2 |
| AttriCLIP | 99.9 | 98.1 | 95.6 | 93.3 | 90.1 | 86.8 | 87.6 | 81.2 | 81.9 | 81.2 | 76.5 | 73.0 | 78.9 | 86.5 |
| SGCL | 99.6 | 97.6 | 96.6 | 93.2 | 90.4 | 86.2 | 86.1 | 85.2 | 84.2 | 82.6 | 80.6 | 80.9 | 81.1 | 88.0 |
| VadCLIP+LWF | 99.8 | 90.5 | 93.5 | 91.3 | 89.8 | 85.2 | 84.4 | 83.8 | 83.0 | 80.3 | 79.8 | 80.3 | 80.6 | 86.3 |
| Continual-CLIP+LWF | 99.5 | 96.4 | 95.6 | 91.0 | 89.2 | 84.5 | 84.6 | 83.8 | 83.2 | 81.4 | 79.1 | 80.2 | 80.1 | 86.8 |
| AttriCLIP+LWF | 99.9 | 97.6 | 95.2 | 91.9 | 89.8 | 85.3 | 85.0 | 83.9 | 82.6 | 81.2 | 80.4 | 80.5 | 80.3 | 87.2 |
| Ours | **99.9** | **98.1** | **97.5** | **95.3** | **92.7** | **88.7** | **88.6** | **87.6** | **86.7** | **84.5** | **83.2** | **83.2** | **83.1** | **89.9** |

Table 2: CL-WSVAD benchmark on XD-Violence in AUC (%) / AP (%). The AUC / AP of Task $i \in \{1, 2, ..., 6\}$, reports the AUC / AP tested over all the previous tasks and the ongoing tasks (*i.e.*, Tasks 1, 2, . . . , $i$). $*$ denotes the reimplemented replay-based continual learning methods.

| Method | Task1 | | Task2 | | Task3 | | Task4 | | Task5 | | Task6 | | Average | |
|---|---|---|---|---|---|---|---|---|---|---|---|---|---|---|
| | AUC | AP | AUC | AP | AUC | AP | AUC | AP | AUC | AP | AUC | AP | AUC | AP |
| LWF | **98.6** | **83.7** | 97.1 | 78.0 | 89.8 | 67.1 | 89.3 | 66.3 | 87.6 | 64.2 | 86.1 | 62.2 | 91.4 | 70.2 |
| DER$^*$ | 98.3 | 82.3 | 96.7 | 75.3 | 90.8 | 68.3 | 90.4 | 67.9 | 91.0 | 71.3 | 90.4 | 72.1 | 92.9 | 72.9 |
| DER++$^*$ | 98.3 | 82.3 | 96.7 | 75.2 | 90.8 | 68.4 | 90.5 | 68.0 | 91.2 | 71.8 | 90.6 | 72.6 | 93.0 | 73.0 |
| Continual-CLIP | 98.4 | 83.5 | 96.8 | 77.6 | 91.7 | 71.1 | 91.9 | 73.4 | 89.0 | 68.3 | 85.7 | 61.4 | 92.3 | 72.5 |
| AttriCLIP | 98.1 | 80.9 | 95.9 | 64.2 | 88.2 | 60.3 | 86.7 | 58.9 | 80.6 | 50.8 | 80.5 | 44.2 | 88.3 | 59.9 |
| SGCL | 98.4 | 83.5 | 96.8 | 77.7 | 91.2 | 70.6 | 91.1 | 71.1 | 88.1 | 66.6 | 86.3 | 62.6 | 92.0 | 72.0 |
| VadCLIP+LWF | 98.4 | 81.9 | 96.9 | 77.9 | 90.0 | 67.8 | 92.1 | 72.2 | 88.9 | 66.5 | 88.8 | 66.0 | 92.5 | 72.1 |
| Continual-CLIP+LW | 98.4 | 83.6 | **97.1** | **78.6** | 90.2 | 69.1 | 89.1 | 67.1 | 87.5 | 65.5 | 86.6 | 64.3 | 91.5 | 71.4 |
| AttriCLIP+LWF | 97.9 | 75.1 | 95.5 | 74.7 | 86.0 | 62.7 | 86.1 | 63.0 | 84.8 | 61.7 | 80.0 | 55.8 | 88.4 | 65.5 |
| Ours | 98.3 | 82.2 | 96.9 | 77.5 | **96.5** | **88.3** | **96.4** | **87.8** | **94.7** | **85.0** | **92.9** | **80.8** | **96.0** | **83.6** |

normal category is also fine-tuned in each subsequent task. The cross-entropy based alignment loss $\mathcal{L}_m = CE(y_{nor}, p_m)$, where $p_m$ represents the video-level prediction derived from the saved video-level features, is utilizing for prompt fine-tuning. The loss function in each subsequent task can be expressed as:

$$\mathcal{L}_{T_i} = \mathcal{L}_{nce} + \mathcal{L}_m + \alpha \mathcal{L}_{tsc}, (i > 1). \tag{9}$$

In addition, NCM is maintained to store core normal features, which serve as representative characteristics distinctly different from anomalies. Meanwhile, learnable prompts are also updated to integrate new anomalies and normal features based on the NCM and newly encountered anomalies. These approaches ensure effective differentiation between normal samples and anomalies that closely resemble normal events in subsequent tasks.

**Update Strategy of Learnable Text Prompt.** As shown in Fig. 1, in Task $i$ ($i > 1$), the learnable text prompt $t_p^i$ for the current anomaly can be updated, while the prompts associated with previously seen anomalies remain frozen to ensure that these seen anomalies are unaffected. In the anomaly continual learning process, only the corresponding learnable text prompt is updated, and the prompts learned in each task are not overwritten. Therefore, this update strategy effectively mitigates catastrophic forgetting. Meanwhile, only a learnable text prompt is updated in one task, effectively reducing computational overhead.

# 4 EXPERIMENTS

## 4.1 DATASETS AND EVALUATION METRICS

**Datasets. UCF-Crime** (Sultani et al., 2018) is a large-scale real-world video dataset for WSVAD task. This dataset involves 13 types of anomalies in surveillance videos, *e.g.*, arson, fighting, rob-

bery, road accident, *etc*. In the continual learning experiments, normal training data are assigned to the initialization task, and the 13 types of abnormal training videos are respectively assigned into initialization task and the remaining 12 tasks. Following existing CL works (Tang et al., 2023; Liu et al., 2024), each anomaly class is assigned to the corresponding task in alphabetical order. In the testing stage, the model trained for each task is evaluated on the testing set using both normal instances and known anomalous videos. **XD-Violence** (Wu et al., 2020), which is a large-scale and multi-scene dataset, possess 4,754 untrimmed videos containing audio signals and video-level labels. XD-Violence, of which source includes movies, cartoons, captured by CCTV cameras, handheld cameras, car driving recorders, etc, contains a total duration of 217 hours. This dataset contains 3,954 training videos and 800 testing videos. XD-Violence provides 6 types of anomalies, and the experimental setup on XD-Violence is consistent with that on UCF-Crime.

**Evaluation Metrics.** Given that WSVAD typically evaluates model detection performance utilizing area under the curve (AUC) or average precision (AP), we develop a benchmark based on AUC or AP to assess the continual learning performance. Following existing CL method (Wang et al., 2023), in Task $i$, we employ the AUC / AP tested over all the previous tasks and the ongoing tasks (*i.e.*, Task 1, 2, ..., i), as the metric.

**Implementation Details.** In our framework, the frozen image and text encoders are based on the pre-trained CLIP (ViT-B/16). Then, we employ a temporal GCN structure (Wu et al., 2020), consisting of two GCN modules with two layers each and one FC layer, to construct the $f_a$. For hyperparameters, $M$ is set to 10 in the learnable text prompts. In Eq. 2, $\tau$ is set to 0.07, and in Eq. 7, $m$ is set to 1. For NCM, the memory size is set to $100 \times 512$ on UCF-Crime and $50 \times 512$ on XD-Violence. Additionally, on UCF-Crime, $\lambda = 1$, $\alpha = 10^{-1}$, and on XD-Violence, $\lambda = 10^{-3}$, $\alpha = 10^{-4}$. Moreover, in the Top-K mean operation, $K = T_n/16 + 1$. In the training stage, we train the model on NVIDIA RTX 3080 GPU by PyTorch, and AdamW (Loshchilov & Hutter, 2017) is utilized as the optimizer. On UCF-Crime, the learning rate is $1 \times 10^{-5}$, with training epochs set to 3 for NRP, 3 for weakly supervised adaptation training, and 10 for the remaining tasks, respectively. On XD-Violence, the learning rate is $2 \times 10^{-5}$, with training epochs set to 1 for NRP, 3 for weakly supervised adaptation training, and 10 for the remaining tasks, respectively.

## 4.2 MAIN RESULTS

In this subsection, we establish the first benchmark for CL-WSVAD on the UCF-Crime and XD-Violence, as detailed in Tab. 1 and Tab. 2. We present the AUC / AP achieved for each task along with their average values, AvgAUC / AvgAP. Note that the AUC / AP illustrates the performance of the current model on seen testing videos, highlighting the model's ability to mitigate catastrophic forgetting, particularly the AUC / AP of the final task. Here, the seen test videos refers to the type of videos in the test set that the model has already encountered in previous or current tasks. Meanwhile, increased AUC / AP values suggest improved performance of the model in mitigating catastrophic forgetting. We employ CoOp as the backbone framework and reimplement continual learning methods, including LWF (Li & Hoiem, 2017), DER (Buzzega et al., 2020), and DER++ (Buzzega et al., 2020) method, for CL-WSVAD. Here, the GCN-based temporal adapter is implemented to the adaptation of CoOp for CL-WSVAD. Meanwhile, we introduce CLIP based continual learning method, including Continual-CLIP (Thengane et al., 2022), AttriCLIP (Wang et al., 2023), SGCL (Yu et al., 2024) for CL-WSVAD. Subsequently, we combine LWF with VadCLIP (Wu et al., 2024b), Continual-CLIP, and AttriCLIP to further evaluate their performance on CL-WSVAD.

It can be observed that LWF, as a regularization-based CL method, exhibits limited performance on CL-WSVAD task. DER and DER++, as replay-based CIL methods, outperform LWF primarily due to the introduction of a minimal number of prior anomalies into subsequent tasks. Although these two methods improve mitigating-forgetting performance, they incur substantial memory, especially in video tasks. Although Continual-CLIP achieves favorable results on UCF-Crime, it performs inadequately in terms of APs on XD-Violence. Then, due to the significant diversity among anomalies, AttriCLIP, which relies on common attribute learning, demonstrates inferior performance, particularly on XD-Violence. SGCL relies on the semantic relationships between previous and subsequent task labels, but the limited text labels and weak correlations among them, resulting in SGCL failing to achieve the anticipated results.

Table 3: Performance comparison across different VAD paradigms on UCF-Crime and XD-Violence.

| Paradigm | Method | UCF-Crime | XD-Violence | |
|---|---|---|---|---|
| | | AUC | AUC | AP |
| UVAD | Conv-AE (Hasan et al., 2016) | 50.60 | - | - |
| | BODS (Wang & Cherian, 2019) | 68.26 | 57.32 | - |
| | GODS (Wang & Cherian, 2019) | 70.46 | 61.56 | - |
| | LANP-UVAD (Shi et al., 2024) | 80.02 | - | - |
| WSVAD | GCNAD (Zhong et al., 2019) | 82.12 | - | - |
| | CLAWS (Zaheer et al., 2020) | 83.03 | - | - |
| | MIST (Feng et al., 2021) | 82.30 | - | - |
| | RTFM (Tian et al., 2021) | 84.03 | - | 77.81 |
| | MSL (Li et al., 2022) | 85.30 | - | 78.28 |
| | BN-SVP (Sapkota & Yu, 2022) | 83.39 | - | - |
| | MGFN (Chen et al., 2023) | 86.98 | - | 79.19 |
| | VadCLIP (Wu et al., 2024b) | 88.02 | - | 84.51 |
| | PE-MIL (Chen et al., 2024) | 86.83 | - | 88.05 |
| | STPrompt (Wu et al., 2024a) | 88.08 | - | - |
| Zero-Shot | ZS CLIP (Radford et al., 2021) | 53.16 | 38.21 | 17.83 |
| | ZS Imagebind (Image) (Girdhar et al., 2023) | 53.65 | 58.81 | 27.25 |
| | ZS Imagebind (Video) (Girdhar et al., 2023) | 55.78 | 55.06 | 25.36 |
| CL-WSVAD | Ours (w/o NCM) | 82.80 | 91.27 | 76.06 |
| | Ours (w/ NCM) | 83.10 | 92.93 | 80.78 |

Without mitigating-forgetting strategies, CLIP based methods tends to overfit to specific sub-tasks in CL-WSVAD. Subsequently, we apply LWF to the current state-of-the-art WSVAD method, VadCLIP, and find that VadCLIP's performance is comparable to that of DER. Next, integrating LWF into Continual-CLIP does not improve performance and, as observed on UCF-Crime, actually restricts Continual-CLIP's effectiveness in new tasks. In contrast, LWF assists AttriCLIP in achieving better performance on UCF-Crime. Finally, our approach achieves the best performance across both datasets, especially on the challenging XD-Violence, without requiring regularization or prior anomaly data.

## 4.3 COMPARISONS WITH VAD PARADIGMS

Here, the CL-WSVAD paradigm is compared with the existing VAD paradigm. As shown in the Tab. 3, we report the anomaly detection performance of the CL-WSVAD model at the final task of continual learning on the entire dataset. Since abnormal videos are not included, the performance of UVAD methods is limited in complex anomaly scenarios. With the introduction of anomalous data, the performance of WSVAD methods is significantly improved. However, given that real-world anomalies are difficult to collect exhaustively and are continuously introduced, the scalability of WSVAD which trained on fixed dataset is limited for streaming anomalies. Upon introducing new anomalies, the WSVAD method requires recalling the previous data and retraining the entire model. Next, the zero-shot (ZS) VAD methods clearly struggle with addressing intricate VAD task. Finally, our CL-WSVAD, which is more aligned with real-world scenarios compared to existing WSVAD paradigms, achieves competitive performance and even surpasses some of the current WSVAD methods. Our method not only enhances the scalability of WSVAD but also improves training efficiency by requiring only minimal parameter updates to the prompts for newly introduced data. Additionally, we integrate the SOTA WSVAD method, VadCLIP, with LWF adapted to the CL-WSVAD task. As shown in the Tab. 1 and Tab. 2, our approach achieves superior results. The results of the zero-short methods are reported by Zanella et al. (2024).

## 4.4 ABLATION STUDY

In this subsection, we conduct an ablation study to evaluate our proposed method. As shown in Tab. 4, we report the AUC / AP achieved in the final task and the AvgAUC / AvgAP. First, we note that the proposed continual learning framework achieves AvgAUC of 86.21% on UCF-Crime, and our framework outperforms LWF and DER in AvgAUC. Additionally, our continual learning framework achieves 73.88% in AvgAP on XD-Violence, surpassing all other methods in AvgAP.

Table 4: Ablation study on UCF-Crime and XD-Violence.

| NRP | MCMA | NCM | UCF-Crime | | XD-Violence | | | |
|:---:|:---:|:---:|:---:|:---:|:---:|:---:|:---:|:---:|
| | | | AUC | AvgAUC | AUC | AvgAUC | AP | AvgAP |
| | | | 78.54 | 86.21 | 87.85 | 93.23 | 66.27 | 73.88 |
| ✓ | | | 80.90 | 86.58 | 90.69 | 94.20 | 73.86 | 77.97 |
| ✓ | ✓ | | 82.80 | 89.69 | 91.27 | 94.93 | 76.06 | 80.19 |
| | ✓ | ✓ | 78.78 | 85.16 | 91.42 | 95.01 | 73.66 | 78.53 |
| ✓ | | ✓ | 81.37 | 88.83 | 92.80 | 95.82 | 77.02 | 80.51 |
| ✓ | ✓ | ✓ | **83.10** | **89.94** | **92.93** | **95.96** | **80.78** | **83.61** |

Table 5: Ablation study for the size of NCM on UCF-Crime.

| Memory Size | 0 | 50×512 | 100×512 | 200×512 | 400×512 |
|:---:|:---:|:---:|:---:|:---:|:---:|
| AUC | 82.80 | 82.79 | 83.10 | 83.14 | 83.17 |
| AvgAUC | 89.69 | 89.71 | 89.94 | 89.95 | 89.99 |

This result effectively validates the continual learning performance of our proposed framework. Then, the introduction of NRP leads to the 2.36% improvement in AUC on UCF-Crime and the 7.59% improvement in AP on XD-Violence. This improvement is primarily due to the learned robust normal representations, which provide the foundation for subsequent anomaly learning. Thereafter, MCMA further enhances performance by 3.11% in AvgAUC on UCF-Crime and 2.22% in AvgAP on XD-Violence. This enhancement primarily results from MCMA guiding the temporal adapter and learnable text prompts to develop more generalized representations of normal and anomalous instances, thereby effectively assisting the pre-trained model in adapting to CL-WSVAD. When NRP or MCMA is removed, the performance of our method declines, particularly in AUC on UCF-Crime and AP on XD-Violence, demonstrating the necessity of NRP and MCMA. Furthermore, the normal video features provided by NCM assist our method in achieving the best performance.

Additionally, we perform ablation experiments to evaluate the impact of the memory size of NCM on UCF-Crime. As shown in Tab. 5, we report the AUC achieved in the final task and AvgAUC. Here, we observe that a memory size of 50×512 is insufficient to enhance model performance. Subsequently, we expand the memory size to 100×512, which results in a 0.3% improvement in AUC and a 0.25% improvement in AvgAUC. Then, the memory is expanded to 200×512 and 400×512, but no significant improvement is observed. Therefore, we set the memory size to 100×512 on UCF-Crime.

## 5 CONCLUSION

In this work, we emphasize that anomalies are streaming in real-world VAD scenarios and pioneer to propose the CL-WSVAD paradigm. Then, we propose a continual learning method to mitigate catastrophic forgetting for WSVAD paradigm. We leverage easily accessible textual anomalies for pre-training, allowing the model to learn a robust normality representation that enhances discrimination between the normality and the increasingly emerging potential anomalies. Next, we propose MCMA method that guides the pre-trained model to effectively adapt to CL-WSVAD. Meanwhile, we propose a continual learning framework which based on retaining the learnable text prompts for each type of anomaly, mitigating catastrophic forgetting. The effectiveness of our method has been demonstrated on the constructed benchmark.

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

# A APPENDIX

## A.1 PROOF OF THE EFFECTIVENESS OF NORMALITY REPRESENTATION PRE-TRAINING

Inspired by Khosla et al. (2020); Oord et al. (2018); Chen et al. (2020); Saunshi et al. (2019), we theoretically demonstrate the effectiveness of NRP in improving the normality visual representation. To evaluate the relevance of video representation to textual semantics feature, mutual information is introduced and defined as follows:

$$I(\boldsymbol{x}, \boldsymbol{w}) = \sum_{\boldsymbol{x}, \boldsymbol{w}} p(\boldsymbol{x}, \boldsymbol{w}) \log \frac{p(\boldsymbol{x}|\boldsymbol{w})}{p(\boldsymbol{x})}, \tag{10}$$

where $\boldsymbol{x} = f_a(\boldsymbol{z})$, and $\boldsymbol{w}$ represents the textual embedding. Here, an increasing value of $I(\boldsymbol{x}, \boldsymbol{w})$ signifies a stronger correlation between the $\boldsymbol{x}$ and $\boldsymbol{w}$ achieved.

Given the challenges in directly estimating $p(\boldsymbol{x}|\boldsymbol{w})$ and $p(\boldsymbol{x})$, following Oord et al. (2018), we introduce a density ratio, $f_I(\boldsymbol{x}, \boldsymbol{w}) \propto \frac{p(\boldsymbol{x}|\boldsymbol{w})}{p(\boldsymbol{x})}$, which preserves the mutual information between $\boldsymbol{x}$ and $\boldsymbol{w}$. Referring to Eq. 2, $f_I$ is defined as $f_I = \exp(sim(\boldsymbol{x}, \boldsymbol{w})/\tau)$ to facilitate the proof process.

In CL-WSVAD, video features set $X$ can be divided into normal video feature set $X_{nor}$, and abnormal video feature set $X_{ab}$. According to Eq. 5, Top-K mean operation is used to obtain the $p_i^v$ for calculating $\mathcal{L}_{nce}$, the $X_{nor}$ consists of the normal video features corresponding to the Top-K video segments. In the NRP process, a normal text embedding $\boldsymbol{w}_0$, along with a set of text embeddings $W_{ab} = \{\boldsymbol{w}_1, \boldsymbol{w}_2, ..., \boldsymbol{w}_{N_t-1}\}$, which obtained by $N_t - 1$ anomalous texts generated by ChatGPT, are applied for pre-training. Combining $f_I = \exp(sim(\boldsymbol{x}, \boldsymbol{w})/\tau)$ with Eq. 2, the $\mathcal{L}_{nce}$ can be expressed as follows:

$$\mathcal{L}_{nce} = - \mathop{\mathbb{E}}_{\boldsymbol{x}_i \in X_{nor}} \left[ \log \frac{\frac{p(\boldsymbol{x}_i|\boldsymbol{w})}{p(\boldsymbol{x}_i)}}{\frac{p(\boldsymbol{x}_i|\boldsymbol{w})}{p(\boldsymbol{x}_i)} + \sum_{\boldsymbol{x}_j \in X_{ab}} \frac{p(\boldsymbol{x}_j|\boldsymbol{w})}{p(\boldsymbol{x}_j)}} \right] \tag{11}$$

**Since there are no abnormal videos in the NRP process, the term** $\sum_{\boldsymbol{x}_j \in X_{ab}} \frac{p(\boldsymbol{x}_j|\boldsymbol{w})}{p(\boldsymbol{x}_j)}$ **does not exist.** Based on the perfect semantic alignment of vision and text features in CLIP, we apply **the generated anomaly text embeddings** $W_{ab}$ **to simulate abnormal video features** $X_{ab}$, and $\mathcal{L}_{nce}$ can be approximated as:

$$\mathcal{L}_{nce} \approx - \mathop{\mathbb{E}}_{\boldsymbol{x}_i \in X_{nor}} \left[ \log \frac{\frac{p(\boldsymbol{x}_i|\boldsymbol{w})}{p(\boldsymbol{x}_i)}}{\frac{p(\boldsymbol{x}_i|\boldsymbol{w})}{p(\boldsymbol{x}_i)} + \sum_{\boldsymbol{w}_j \in W_{ab}} \frac{p(\boldsymbol{w}_j|\boldsymbol{w})}{p(\boldsymbol{w}_j)}} \right] \tag{12}$$

$$= \mathop{\mathbb{E}}_{\boldsymbol{x}_i \in X_{nor}} \log \left[ 1 + \frac{p(\boldsymbol{x}_i)}{p(\boldsymbol{x}_i|\boldsymbol{w})} \sum_{\boldsymbol{w}_j \in W_{ab}} \frac{p(\boldsymbol{w}_j|\boldsymbol{w})}{p(\boldsymbol{w}_j)} \right] \tag{13}$$

$$\approx \mathop{\mathbb{E}}_{\boldsymbol{x}_i \in X_{nor}} \log \left[ 1 + \frac{p(\boldsymbol{x}_i)}{p(\boldsymbol{x}_i|\boldsymbol{w})} (N_t - 1) \mathop{\mathbb{E}}_{\boldsymbol{w}_j} \frac{p(\boldsymbol{w}_j|\boldsymbol{w})}{p(\boldsymbol{w}_j)} \right] \tag{14}$$

$$= \mathop{\mathbb{E}}_{\boldsymbol{x}_i \in X_{nor}} \log \left[ 1 + \frac{p(\boldsymbol{x}_i)}{p(\boldsymbol{x}_i|\boldsymbol{w})} (N_t - 1) \right] \tag{15}$$

$$\geq \mathop{\mathbb{E}}_{\boldsymbol{x}_i \in X_{nor}} \log \left[ \frac{p(\boldsymbol{x}_i)}{p(\boldsymbol{x}_i|\boldsymbol{w})} N_t \right] \tag{16}$$

$$= -I(\boldsymbol{x}, \boldsymbol{w}) + \log(N_t). \tag{17}$$

As we known, $I(\boldsymbol{x}, \boldsymbol{w})$, where $\boldsymbol{x} \in X_{nor}$, represents the mutual information between the normal video representation and text embeddings. Next, the following inequality relationship is obtained:

$$I(\boldsymbol{x}, \boldsymbol{w}) \geq \log(N_t) - \mathcal{L}_{nce}, \tag{18}$$

Table 6: Experiments in different continual learning configurations on UCF-Crime in AUC (%).

| Configuration | Task1 | Task2 | Task3 | Task4 | Task5 | Task6 | Task7 | Task8 | Task9 | Task10 | Task11 | Task12 | Task13 |
|---|---|---|---|---|---|---|---|---|---|---|---|---|---|
| Config.1 | 99.9 | 98.1 | 97.5 | 95.3 | 92.7 | 88.7 | 88.6 | 87.6 | 86.7 | 84.5 | 83.2 | 83.2 | 83.1 |
| Config.2 | 99.9 | 97.9 | 97.5 | 94.7 | 93.0 | 88.6 | 88.4 | 87.5 | 87.7 | 85.5 | 84.1 | 83.7 | 83.5 |

Table 7: Experiments in different continual learning configurations on XD-Violence in AUC (%) / AP (%).

| Configuration | Task1 | | Task2 | | Task3 | | Task4 | | Task5 | | Task6 | |
|---|---|---|---|---|---|---|---|---|---|---|---|---|
| | AUC | AP | AUC | AP | AUC | AP | AUC | AP | AUC | AP | AUC | AP |
| Config.1 | 98.3 | 82.2 | 96.9 | 77.5 | 96.5 | 88.3 | 96.4 | 87.8 | 94.7 | 85.0 | 92.9 | 80.8 |
| Config.2 | 90.6 | 63.4 | 90.8 | 62.7 | 92.7 | 84.1 | 95.0 | 86.9 | 92.4 | 83.2 | 90.5 | 80.0 |

where the lower bound of the $I(\boldsymbol{x}, \boldsymbol{w})$ can be derived. In the NRP process, as $\mathcal{L}_{nce}$ decreases, the lower bound of $I(\boldsymbol{x}, \boldsymbol{w})$ increases continuously. Moreover, the introduction of abnormal textual information generated by ChatGPT increases the overall sample size $N_t$, which simultaneously raises the lower bound of $I(\boldsymbol{x}, \boldsymbol{w})$. As the lower bound of $I(\boldsymbol{x}, \boldsymbol{w})$ is increased, it indicates that the normal video representation and the normal text embedding have achieved a stronger correlation, thereby validating the effectiveness of our method in enhancing the normal video representation. In fact, in the experiment in Appendix A.4, we also experimentally validate that with the increasing introduction of abnormal texts, our method achieves better performance.

## A.2 EXPERIMENTS IN ANOTHER CONTINUAL LEARNING CONFIGURATION

For the experimental configuration outlined in the main text, designated as Config.1, normal videos are provided all at once in the Task 1, with only new anomalies introduced in each subsequent continual learning task. Considering the hardship of collecting all normal videos during the initial learning stage, there is another experimental configuration: to guide the model in simultaneously learning normal and anomalous patterns, a comparable number of normal and anomalous samples are introduced concurrently in each continual learning task. This configuration is designated as Config.2. Here, we implement our method under Config.2. In the testing stage, both configurations utilize the same test data, which includes all normal videos and known abnormal videos, and same metrics for each continual learning task, with the results on the two datasets presented in Tab. 6 and Tab. 7.

It can be observed that under different configurations, our method achieves comparable performance in the final task. This validates that our approach can sufficiently leverage normal videos to effectively achieve the expected results of the CL-WSVAD task in both configurations. Moreover, in Config.1, the initialization task introduces all normal data, leading to better results in the earlier continual learning tasks, with this observation being particularly evident in the XD-Violence dataset. It is essential to note that in the testing stage of each continual learning task, all normal videos from the testing set are utilized. Consequently, the Config.1, which employs more normal data for training in Task 1, outperforms Config.2 in the earlier tasks. In addition, compared to the normal videos primarily sourced from simple surveillance scenes on UCF-Crime, the normal videos in XD-Violence, derived from movies and YouTube, exhibit greater diversity. As a result, the performance differences across different configurations are more pronounced on XD-Violence.

## A.3 CONTINUAL LEARNING EXPERIMENTS WITH MULTI-CLASS INCREMENTAL CONFIGURATION

In Tab. 1 and Tab. 2, the experiments focus on the continual learning task containing only one anomaly type. Additionally, we evaluate a multi-class incremental configuration, where multiple anomaly types (2, 4, or 6 types) are sequentially introduced. The results, as shown in Tab. 8, demon-

Table 8: Continual learning experiments with multi-class incremental configuration on UCF-Crime in AUC (%). Num represents the number of anomaly types introduced in each task.

| Num | Task1 | Task2 | Task3 | Task4 | Task5 | Task6 | Task7 | Task8 | Task9 | Task10 | Task11 | Task12 | Task13 |
|-----|-------|-------|-------|-------|-------|-------|-------|-------|-------|--------|--------|--------|--------|
| 1 | 99.9 | 98.1 | 97.5 | 95.3 | 92.7 | 88.7 | 88.6 | 87.6 | 86.7 | 84.5 | 83.2 | 83.2 | 83.1 |
| 2 | - | 97.7 | - | 95.6 | - | 89.4 | - | 88.1 | - | 84.7 | - | 83.6 | 83.6 |
| 4 | - | - | - | 96.4 | - | - | - | 89.7 | - | - | - | 84.9 | 84.8 |
| 6 | - | - | - | - | - | 91.2 | - | - | - | - | - | 86.3 | 86.1 |

strate improved performance as more classes are introduced at once, highlighting the generalization capability of our approach for both single- and multi-class continual learning.

### A.4 ANALYSIS OF ANOMALOUS TEXTS IN NRP

To further validate the effectiveness of NRP, we have separately analyzed the impact of the number of and content of the anomalous texts on performance.

Table 9: Analysis of the number of anomaly texts in the NRP on UCF-Crime.

| Num | 0 | 100 | 500 | 1000 | 2000 | 4000 |
|-----|-----|-----|-----|------|------|------|
| AUC | 78.78 | 80.94 | 82.73 | 82.78 | 83.10 | 82.79 |
| AvgAUC | 85.16 | 87.55 | 89.53 | 89.50 | 89.94 | 89.55 |

We conduct ablation study on the number of potential anomaly texts used in NRP. As shown in Tab. 9, we report the AUC achieved in the final task and AvgAUC, and the results show progressively improved performance with an increasing number of anomaly texts. Even learning with only 100 anomaly texts yields a significant improvement (+2% in AUC), with 2,000 texts resulting in the best performance. However, due to limitations in ChatGPT's generation capabilities, some irrelevant anomalies are present among the excessive samples, which limit further improvement when applying 4,000 potential anomaly texts.

Table 10: Analysis of the content of anomalous texts in the NRP.

| | UCF-Crime | | XD-Violence | | | |
|---|-----|-----|-----|-----|-----|-----|
| | AUC | AvgAUC | AUC | AvgAUC | AP | AvgAP |
| w/o Relevant Anomalous Texts | 82.97 | 89.64 | 93.28 | 96.10 | 80.46 | 82.92 |
| w Relevant Anomalous Texts | 83.10 | 89.94 | 92.93 | 95.96 | 80.78 | 83.61 |

To avoid information leakage, we do not leverage any information related to video/image or anomaly categories when guiding ChatGPT to generate anomaly texts. These generated anomaly items are dataset-agnostic, meaning we could use the same set of anomaly items for both UCF-Crime and XD-Violence. To gain a deeper insight into how these anomaly texts affect detection, we remove the anomaly items related to the specific anomaly categories in both datasets, approximately 240 items out of the 2,000. As shown in Tab. 10, we report the AUC achieved in the final task and AvgAUC, and the results show that these removed items have a negligible impact on performance.

### A.5 PERFORMANCE VALIDATION IN ADDITIONAL METRICS

Here, we separately evaluate AUC scores on the current task in CAUC and previous tasks in PAUC to validate the model's ability to mitigate catastrophic forgetting. The PAUC of Task $i \in \{1, 2, ..., 13\}$, reports the AUC tested over all the previous tasks (*i.e.*, Tasks 1, 2, . . . , $i-1$), while the CAUC of Task $i \in \{1, 2, ..., 13\}$, reports the AUC tested on the current tasks (Tasks $i$). Results on UCF-Crime

Table 11: Performance validation in additional metrics on UCF-Crime in PAUC (%) and CAUC (%).

| Method | Task1 | | Task2 | | Task3 | | Task4 | | Task5 | | Task6 | | Task7 | |
|---|---|---|---|---|---|---|---|---|---|---|---|---|---|---|
| | PAUC | CAUC | PAUC | CAUC | PAUC | CAUC | PAUC | CAUC | PAUC | CAUC | PAUC | CAUC | PAUC | CAUC |
| DER++* | - | 99.7 | 99.7 | 93.0 | 92.3 | 97.0 | 93.8 | 87.3 | 89.3 | 91.1 | 87.6 | 79.9 | 82.6 | 94.4 |
| SGCL | - | 99.6 | 99.9 | 97.7 | 97.1 | 98.2 | 96.6 | 91.9 | 92.8 | 93.4 | 90.3 | 86.8 | 86.2 | 96.6 |
| VadCLIP+LWF | - | 99.8 | 99.2 | 90.5 | 91.3 | 97.6 | 94.7 | 91.2 | 91.9 | 94.1 | 89.4 | 82.9 | 84.8 | 91.4 |
| Continual-CLIP+LWF | - | 99.5 | 98.3 | 96.4 | 96.6 | 96.8 | 95.4 | 87.8 | 91.1 | 92.7 | 89.0 | 77.5 | 84.4 | 96.5 |
| Ours | - | 99.9 | 99.9 | 98.3 | 98.2 | 98.6 | 97.6 | 94.4 | 95.1 | 94.4 | 92.7 | 90.7 | 88.6 | 98.5 |

| Method | Task8 | | Task9 | | Task10 | | Task11 | | Task12 | | Task13 | | Average | |
|---|---|---|---|---|---|---|---|---|---|---|---|---|---|---|
| | PAUC | CAUC | PAUC | CAUC | PAUC | CAUC | PAUC | CAUC | PAUC | CAUC | PAUC | CAUC | PAUC | CAUC |
| DER++* | 83.4 | 90.4 | 84.3 | 82.0 | 82.2 | 92.0 | 80.7 | 83.9 | 81.0 | 98.9 | 81.2 | 95.3 | 86.5 | 91.2 |
| SGCL | 85.9 | 88.7 | 85.3 | 82.3 | 84.0 | 93.1 | 82.4 | 79.3 | 80.6 | 98.0 | 81.2 | 91.5 | 88.5 | 92.1 |
| VadCLIP+LWF | 84.6 | 95.2 | 83.9 | 86.2 | 82.7 | 90.1 | 80.7 | 84.2 | 79.7 | 98.5 | 80.9 | 95.7 | 87.0 | 92.1 |
| Continual-CLIP+LWF | 84.4 | 86.3 | 83.9 | 85.5 | 83.1 | 92.1 | 81.3 | 75.5 | 79.5 | 99.0 | 80.3 | 87.4 | 87.3 | 90.2 |
| Ours | 88.5 | 93.6 | 87.7 | 77.9 | 86.5 | 93.4 | 84.4 | 88.1 | 83.0 | 99.0 | 83.2 | 97.0 | 90.5 | 94.1 |

Table 12: Analysis of the hyperparameters of the loss function.

| | UCF-Crime | | | | XD-Violence | | |
|---|---|---|---|---|---|---|---|
| $\lambda$ | 0.1 | 1 | 10 | $\lambda$ | 0.01 | 0.001 | 0.0001 |
| AUC | 82.76 | 83.10 | 81.49 | AP | 78.15 | 80.78 | 78.08 |
| AvgAUC | 89.81 | 89.94 | 88.61 | AvgAP | 82.07 | 83.61 | 81.12 |
| $\alpha$ | 1 | 0.1 | 0.01 | $\alpha$ | $10^{-3}$ | $10^{-4}$ | $10^{-5}$ |
| AUC | 81.43 | 83.10 | 79.75 | AP | 80.43 | 80.78 | 80.40 |
| AvgAUC | 87.74 | 89.94 | 87.13 | AvgAP | 83.54 | 83.61 | 83.39 |

in Tab. 11 demonstrate our superiority in both PAUC and CAUC, indicating that our method effectively learns newly introduced anomalies while maintains the ability to detect previously observed anomalies. Meanwhile, the performance trends of PAUC and CAUC are nearly consistent with those provided in Tab. 1, indicating that the evaluation metric in the main text is sufficient to validate the performance of our method.

## A.6    ANALYSIS OF THE HYPERPARAMETERS OF THE LOSS FUNCTION

Due to the distinct data domains—XD-Violence consists of movies and YouTube videos, while UCF-Crime features surveillance footage—both normal and abnormal videos from these datasets exhibit varying levels of diversity, leading to different optimal values for hyperparameters across datasets. Here, we additionally provide ablation studies for $\alpha$ and $\lambda$ on UCF-Crime and XD-Violence. As shown in Tab. 12, where we report the AUC achieved in the final task and AvgAUC, the hyperparameter values we apply achieve the best results.

## A.7    PERFORMANCE VALIDATION ON CROSS-DATASET EXPERIMENTS

Table 13: Performance validation on cross-dataset experiments in AUC (%).

| Method | UCF-Crime | XD-Violence | | | | | |
|---|---|---|---|---|---|---|---|
| | Task13 | Task14 | Task15 | Task16 | Task17 | Task18 | Task19 |
| VadCLIP+LWF | 80.62 | 72.24 | 71.53 | 75.38 | 75.36 | 74.26 | 73.42 |
| SGCL | 81.07 | 82.32 | 82.38 | 85.68 | 85.66 | 84.32 | 82.58 |
| Continual-CLIP+LWF | 80.13 | 82.75 | 83.12 | 84.79 | 84.93 | 83.82 | 82.86 |
| Ours | **83.10** | **82.79** | **83.18** | **88.52** | **88.45** | **87.10** | **85.78** |

To further validate the scalability of our method, we construct a larger-scale benchmark by combining UCF-Crime and XD-Violence. Specifically, each anomaly type from XD-Violence is sequentially appended to UCF-Crime in an incremental process. As shown in the Tab. 13, our method ef-

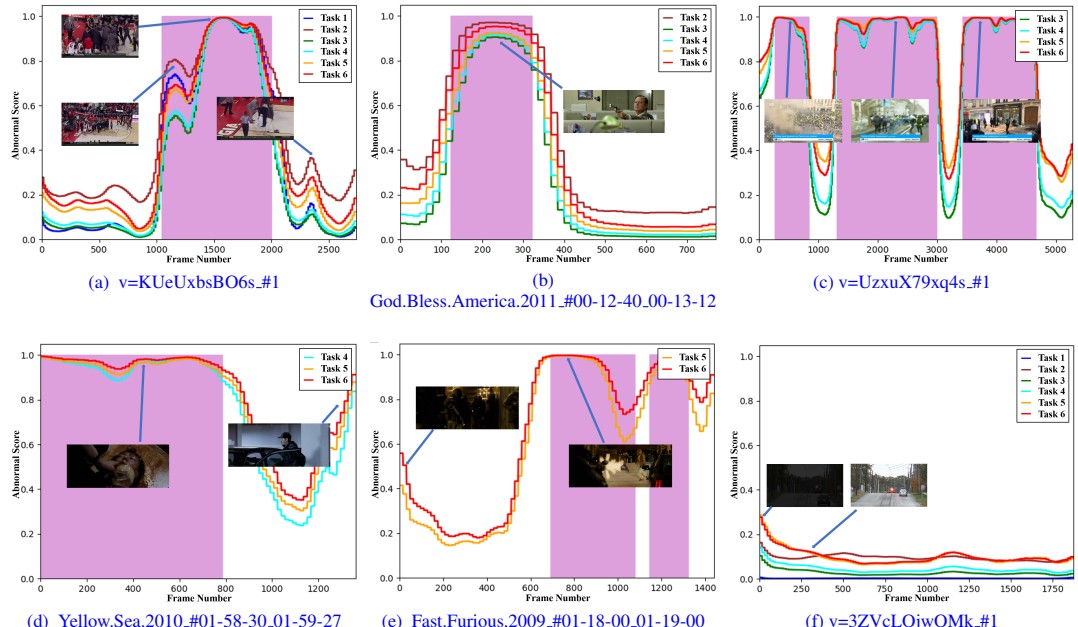

Figure 2: Qualitative results on XD-Violence. The horizontal axis represents the frame number in the temporal sequence, while the vertical axis represents the anomaly scores. The plum-colored columns correspond to the ground-truth abnormal regions.

fectively addresses the cross-dataset setting and generalizes well to an increasing number of anomaly types.

## A.8 ANALYSIS FOR THE TASK ORDER

Table 14: Analysis for the task order on UCF-Crime.

| Sequence ID | S1 | S2 | S3 | S4 | S5 | Average |
|---|---|---|---|---|---|---|
| AUC (%) | 82.84 | 83.45 | 82.61 | 83.50 | 83.66 | 83.21 |

Here, we further analyze the impact of task order on the results of continual learning. Specifically, we randomly shuffle the original order of introduced anomaly types and randomly select 5 different shuffled task sequences. As shown in the Tab. 14, we list the AUC achieved on the final task on UCF-Crime in each sequence.

The results obtained on the randomly shuffled sequences are close to the result we achieved on the original task order, where AUC=83.10% on UCF-Crime. It can be observed that the task order does not significantly impact our experimental results, as our method achieves favorable performance across multiple randomly shuffled sequences.

## A.9 QUALITATIVE ANALYSES

Here, we analyze the effectiveness of our proposed method based on visualization results. As shown in Fig. 2, we visualize the predictions of the videos in their corresponding tasks. Since the anomalous videos in Task 6 are not introduced in the previous tasks, we do not apply them to demonstrate the effectiveness. Clearly, the predictions between each subsequent task are comparable, effectively validating the performance of our method in mitigating catastrophic forgetting. Additionally, while our method demonstrates promising performance, it causes false alarms for some hard cases, such as the rapid scene transitions in Fig. 2 (d) and the person holding a gun in Fig. 2 (e).

### A.10 LIMITATION OF OUR METHOD

In the CL-WSVAD paradigm, anomalies are not directly visible to one another, which may limit the overall performance of anomaly detection. Incorporating more comprehensive and diverse prior knowledge about anomalies into the model could be a promising direction for further improving detection performance.

### A.11 MORE IMPLEMENTATION DETAILS FOR THE COMPARED METHODS

In this section, we provide more re-implemented details for comparison methods.

**LWF.** Since CoOp (Zhou et al., 2022) employs learnable text prompts for training, which is similar with our continual learning framework, we apply CoOp as the backbone for this experiment. Here, CoOp uses the same number of learnable parameters as our method, and a GCN-based temporal adapter is employed to adapt CoOp to the CL-WSVAD task. Following the LWF approach, the knowledge distillation loss, which is effective in encouraging the outputs of one network to approximate those of another, is introduced as the training loss. This loss fuction can be expressed as:

$$\mathcal{L}_{old}(y_o, \hat{y}_o) = -\sum_{i=1}^{l} y_o^{'(i)} \log \hat{y}_o^{'(i)}, \tag{19}$$

where $l$ is the number of labels, and $y_o^{'(i)}$, $\hat{y}_o^{'(i)}$ are the modified versions of recorded and current probabilities. They can be represented as:

$$y_o^{'(i)} = \frac{\left(y_o^{(i)}\right)^{1/T}}{\sum_j \left(y_o^{(j)}\right)^{1/T}}, \hat{y}_o^{'(i)} = \frac{\left(\hat{y}_o^{(i)}\right)^{1/T}}{\sum_j \left(\hat{y}_o^{(j)}\right)^{1/T}}, \tag{20}$$

where the $T$ is set to 2 on both UCF-Crime and XD-Violence. The overall loss in the training process can be expressed as:

$$\mathcal{L}_{LWF} = \gamma_1 \mathcal{L}_{old} + \mathcal{L}_{nce} + \alpha \mathcal{L}_{tsc}, \tag{21}$$

where $\mathcal{L}_{tsc}$ is the textual semantic contrastive loss, and $\mathcal{L}_{nce}$ is the cross-entropy loss for the current task. Additionally, the setting of the hyperparameter $\alpha$ is consistent with that of our method. Moreover, the hyperparameter $\gamma_1$ is set to 0.01 on the UCF-Crime dataset and 1 on the XD-Violence. On both UCF-Crime and XD-Violence, we employ the same optimizer, training epoch, and learning rate as other methods.

**DER.** DER (Buzzega et al., 2020) is an effective replay-based continual learning method. Here, we apply CoOp as the backbone and employ a GCN-based temporal adapter to adapt CoOp to the CL-WSVAD task. The replay loss for DER can be represented as follows:

$$\mathcal{L}_{d1} = \|P_r - f_a(Z_r)\|_2, \tag{22}$$

where $Z_r$ represents the stored inputs from previous tasks, $P_r$ denotes the corresponding output of $f_a$ obtained by $Z_r$ on previous tasks. Note that $f_a$ is the trained adapter on the current task. The overall loss in the training process can be expressed as:

$$\mathcal{L}_{DER} = \gamma_2 \mathcal{L}_{d1} + \mathcal{L}_{nce} + \alpha \mathcal{L}_{tsc}, \tag{23}$$

where the hyperparameter $\gamma_2$ is set to 0.01 on both the UCF-Crime dataset and XD-Violence. In the first task, we save a mini-batch of training data for replay. For each subsequent task, we retain 10% of the training data for each type of anomaly for replay. During the training process, the optimizer and learning rate remain consistent with other methods.

**DER++.** DER++ (Buzzega et al., 2020) is an improved version of DER, achieving better performance in mitigating forgetting. We also apply CoOp with the same settings as the backbone and employ a GCN-based temporal adapter to adapt CoOp to the CL-WSVAD task. DER++ retains ground truth labels for replay, and the additional loss can be formulated as follows:

$$\mathcal{L}_{d2} = BCE(y_r, S_a), \tag{24}$$

where $S_a$ denotes the anomaly score obtained from the stored inputs $Z_r$ by the current model, and $y_r$ represents the ground truth label corresponding to $Z_r$. The overall loss in the training process can be expressed as:

$$\mathcal{L}_{DER++} = \gamma_2 \mathcal{L}_{d1} + \gamma_3 \mathcal{L}_{d2} + \mathcal{L}_{nce} + \alpha \mathcal{L}_{tsc}, \tag{25}$$

where the hyperparameter $\gamma_3$ is set to 0.1 on the UCF-Crime dataset and 0.01 on the XD-Violence. The other training settings for DER++ remain consistent with those of DER.

**Continual-CLIP.** Here, we append a GCN-based temporal adapter after the image encoder of Continual-CLIP (Thengane et al., 2022) to adapt it to the CL-WSVAD task. The $\mathcal{L}_{nce}$ is utilized as the loss function for training. Meanwhile, we employ the same optimizer and learning rate as other methods.

**AttriCLIP.** Then, we utilize the GCN-based temporal adapter to guide AttriCLIP (Wang et al., 2023) in adapting to the CL-WSVAD task. We utilize the same prompt length, number of attributes in the bank, and top-C settings as AttriCLIP. The matching loss adopted to optimize the keys can be expressed as:

$$\mathcal{L}_k = \sum_{i=1}^{C} sim(\boldsymbol{z_j}, \boldsymbol{k_{j_i}}), \tag{26}$$

where $\boldsymbol{z_j}$ is the image embedding from the CLIP image encoder, and $\boldsymbol{k_{j_i}}$ denotes on of the top-$C$ keys selected from keys specifically for the $j-th$ image. Note that $sim(\cdot, \cdot)$ is the cosine similarity. Then, the loss to orthogonalize the embeddings of different prompts to increase the diversity of the prompts can be expressed as:

$$\mathcal{L}_p = \frac{1}{N_c(N_c - 1)} \sum_{i=1}^{N_c} \sum_{j=i+1}^{N_c} sim(\boldsymbol{w_i}, \boldsymbol{w_j}), \tag{27}$$

where $N_c$ represents the total number of all classes, and $\boldsymbol{w_i}$ is the text embedding. The loss function in the training process can be expressed as follows:

$$\mathcal{L}_{AC} = \gamma_4 \mathcal{L}_k + \mathcal{L}_p + \mathcal{L}_{nce} + \alpha \mathcal{L}_{tsc}, \tag{28}$$

where $\gamma_4$ is 0.01 on both UCF-Crime and XD-Violence. For the training settings, we employ the same optimizer, training epochs, and learning rate as other methods.

**VadCLIP+LWF.** VadCLIP (Wu et al., 2024b) is the state-of-the-art method for WSVAD, and we introduce LWF into VadCLIP as a continual learning method for comparison. Specifically, we maintain the model architecture, parameter settings, and loss function of the VadCLIP method, and the loss function for the WSVAD task can be expressed as $\mathcal{L}_{ws}$. Therefore, The loss function on training process can be expressed as follows:

$$\mathcal{L}_{VadC} = \mathcal{L}_{ws} + \gamma_5 \mathcal{L}_{old}, \tag{29}$$

where $\gamma_5$ is set to 1 on UCF-Crime and 0.1 on XD-Violence. Here, we still maintain the same training settings as other methods.

**Continual-CLIP+LWF.** Then, we introduce LWF into the continual learning method, Continual-CLIP, in an attempt to achieve better performance as a comparative method for our approach. The loss function in the training process can be expressed as:

$$\mathcal{L}_{CCL} = \mathcal{L}_{CC} + \gamma_6 \mathcal{L}_{old}, \tag{30}$$

where $\gamma_6$ is set to 1 both on UCF-Crime and XD-Violence. We maintain the same model architecture and parameter settings as Continual-CLIP, and in the training stage, we keep the same optimizer and learning rate as that of LWF.

**AttriCLIP+LWF.** Since AttriCLIP does not achieve the desired performance on CL-WSVAD, we introduce LWF into AttriCLIP in an attempt to improve its performance on CL-WSVAD as a comparative method for our approach. We maintain the model architecture and parameter settings of AttriCLIP, and the loss function in the training process can be expressed as:

$$\mathcal{L}_{ACL} = \mathcal{L}_{AC} + \gamma_7 \mathcal{L}_{old}, \tag{31}$$

where $\gamma_7$ is set to 1 on UCF-Crime and 10 on XD-Violence. Here, we still set the optimizer, learning rate, and number of training epochs consistent with those used in other methods.

**SGCL.** SGCL (Yu et al., 2024) proposes that the semantic knowledge contained in the label information provides important semantic cues, which can be linked to previously acquired knowledge of semantic classes. Based on the CLIP model, SGCL introduces this semantic knowledge into continual learning and designs a continual learning method based on the CLIP model. To adapt SGCL to the CL-WSVAD task, we introduce the GCN-based temporal adapter after the CLIP image encoder. The loss function in the training stage can be formulated as follows:

$$\mathcal{L}_{SGCL} = \mathcal{L}_{nce} + \gamma_8 \mathcal{L}_{SG-RL} + \gamma_9 \mathcal{L}_{SG-KD}, \tag{32}$$

where $\mathcal{L}_{SG-RL}$ and $\mathcal{L}_{SG-KD}$ represent the loss functions for intra-task semantically-guided representation learning and inter-task semantically-guided knowledge distillation, respectively. Here, $\gamma_8$ is set to 0.5 for UCF-Crime and 0.1 for XD-Violence, and $\gamma_9$ is set to 0.1 for both datasets. In addition, the same optimizer and training parameter settings as other methods are used. To ensure a fair comparison with other CLIP-based continual learning methods, such as Continual-CLIP and AttriCLIP, we do not adopt the rehearsal strategy of SGCL.

