# OpenReview forum: "Anomalies are Streaming: Continual Learning for Weakly Supervised Video Anomaly Detection"
_ICLR.cc/2025/Conference — Submitted to ICLR 2025_

### Official Review · Reviewer_oFsJ · 2024-10-29

**Soundness:** 4
**Presentation:** 3
**Contribution:** 2
**Rating:** 5
**Confidence:** 5

**Summary:**

In this paper, the authors explore weakly-supervised video anomaly detection in a continual learning paradigm, which the authors believe meets the needs of real-world scenarios. In accordance with the proposed paradigm, the authors define a new setting of this task, which involves multiple periods. In the first period, the model is provided with both normal videos and abnormal videos of a specific type. In the following periods, the model is afforded abnormal videos of exclusive types. Correspondingly, the authors devise a mixed-up cross-modal alignment method to address this setting of VAD. In the first "task", the authors first learn normal patterns by normality representation learning and learn anomalies of the first type by weakly supervised adaptation. In the following tasks, the authors propose anomaly continual learning to better distinguish new anomalies.

**Strengths:**

1. The authors apply themselves to explore a more realistic scenario of video anomaly detection where novel anomaly types keep coming as the detection proceeds.
2. The proposed method is appropriate in solving the challenges of the task and extensive experiments prove the effectiveness of proposed method.
3. The motivation is solid and the overall writing is clear and easy to follow.

**Weaknesses:**

The proposed framework of continual learning for weakly supervised video anomaly detection (CL-WSVAD) presents significant limitations that may hinder its generalization in real-world applications. Concerns arise from several key aspects. Firstly, CL-WSVAD operates under the assumption that all normal videos are collected during the initial stage. However, in practical scenarios, it is often infeasible to gather all normal samples simultaneously, as new unseen normal videos are continuously introduced. Secondly, CL-WSVAD posits that only a single specific type of anomaly will be introduced in subsequent stages. This assumption is overly idealistic, as it does not account for the unpredictability of future anomalies. In reality, annotators cannot reliably specify the types of anomalies that may arise, necessitating the identification or meticulous categorization of anomalies, which poses an additional burden.

Moreover, the performance of the CL-WSVAD method appears uncompetitive compared to existing weakly supervised approaches, raising questions regarding the advantages of the proposed framework. Given that the model could simply be retrained using the complete dataset, the necessity for continual learning seems redundant and lacking in substantial benefits.

Additionally, while the authors introduce a method resembling Contrastive Predictive Coding (CPC) and provide a theoretical proof in Appendix A.1, the rigor of this proof is insufficient. This inadequacy undermines the theoretical support for the proposed method.

**Questions:**

Settings:

The authors should provide a more detailed justification for the rationale and advantages of the proposed setting in comparison to the latest unsupervised video anomaly detection (UVAD) and weakly supervised video anomaly detection (WSVAD) methods, particularly in terms of performance, computational overhead, and the challenges associated with data collection.

Introduction:

1. In lines 44-46, the authors describe WSVAD method in two stages. I believe such a description is uncommon and misses the essence of WSVAD. The author should polish this phase to reveal the typical idea of multi-instance learning in WSVAD.
2. To emphasize the advantages of CL-WSVAD, it is recommended that the author should depict the strength when handling novel scenes comparing to UVAD and WSVAD in introduction. It will stress the practical value of CL-WSVAD.

Experiments:

The authors lack performance comparison with the latest UVAD and WSVAD methods in 2024. Such comparisons are critical for evaluating the proposed setting and method's effectiveness relative to SOTA UVAD and WSVAD methods.

Appendix:

The proof in Appendix A.1 is unreliable. The notation of Eq. 11, e.g., t_(l,n), x_n, is not presented in Eq. 5, which causes confusion of readers.  A more through theoretical proof should be provided.

Typos:

1. In lines 108, 115, 118, 119, and 120, phase like Wu et al.(Wu et al., 2020), should be Wu et al., (2020), using latex command \citet, instead of \citetp, according to the formatting instructions (line 85-86).
2. In lines 263 and 312, the abbreviation for equation will be more appropriate to be Eq., rather than Equ..

---

### Official Review · Reviewer_JakS · 2024-11-02

**Soundness:** 3
**Presentation:** 3
**Contribution:** 3
**Rating:** 5
**Confidence:** 5

**Summary:**

This paper pioneers the exploration of continual learning for WSVAD to address catastrophic forgetting when detecting new anomalies. The authors introduce a normality representation pre-training phase, utilizing potential anomaly texts to enhance the model's ability to learn robust normality representations, thereby improving discrimination against incremental anomalies. Additionally, a mixed-up cross-modal alignment method is proposed to aid in adapting the pre-trained model for CL-WSVAD. The continual learning framework sequentially retains learnable text prompts for each anomaly type, effectively mitigating catastrophic forgetting.

**Strengths:**

1.	The authors pioneer to explore continual learning in WSVAD, which is an inspiring work to study more realistic setting of VAD.
2.	The employment method of prompt learning is innovative and efficient.

**Weaknesses:**

1.	The setting of CL-WSVAD is not solid. As both normal and abnormal videos will appear as the system running continually. Thus, the setting of exclusive anomaly types is task is i is not practical in real scenarios.
2.	The hardship of collecting all normal videos in the initialization learning and exclusive anomaly types in subsequent tasks is not considered, which requires a huge resource to collect and annotate.
3.	The proof of proposed CPC-like method is not solid.

**Questions:**

Settings:
The authors should further elaborate on the rationale and advantages of their proposed setting in comparison to UVAD and original WSVAD. Specifically, they should discuss how their approach performs in terms of accuracy, robustness, and adaptability to unseen anomalies. Additionally, it would be beneficial to analyze the computational overhead associated with their method and hardship of collecting data in real scenarios versus that of UVAD and WSVAD techniques. By providing a thorough argument that highlights these aspects, the authors can effectively justify the relevance and superiority of their approach within the context of contemporary anomaly detection methods.
Experiments:
The authors do not provide a performance comparison with the latest unsupervised video anomaly detection (UVAD) [1] and weakly supervised video anomaly detection (WSVAD) [2, 3] methods introduced in 2024. This omission is significant, as comparing the proposed approach against state-of-the-art techniques is crucial for evaluating its effectiveness of the proposed setting.
[1] Zhang, Menghao, et al. "Multi-Scale Video Anomaly Detection by Multi-Grained Spatio-Temporal Representation Learning." Proceedings of the IEEE/CVF Conference on Computer Vision and Pattern Recognition. 2024.
[2] Chen, Junxi, et al. "Prompt-Enhanced Multiple Instance Learning for Weakly Supervised Video Anomaly Detection." Proceedings of the IEEE/CVF Conference on Computer Vision and Pattern Recognition. 2024
[3] Jain, Yashika, Ali Dabouei, and Min Xu. "Cross-Domain Learning for Video Anomaly Detection with Limited Supervision." arXiv preprint arXiv:2408.05191 (2024).
Proof in Appendix:
The proof in Appendix is not elaborate and confusing, as the proof introduces new notations, which doesn’t appear in early section. The equation is largely similar to proof in CPC, however the rationality of such proof is not solid. The authors should revise to make the proof more strict.

---

### Official Review · Reviewer_zzoQ · 2024-11-04

**Soundness:** 3
**Presentation:** 3
**Contribution:** 3
**Rating:** 6
**Confidence:** 4

**Summary:**

This paper presents a novel approach for continual learning in weakly supervised video anomaly detection (CL-WSVAD) to address the challenge of catastrophic forgetting as new anomalies are introduced over time.

**Strengths:**

The paper introduces a novel approach to address continual learning for weakly supervised video anomaly detection, which is relevant for real-world applications.
The normality representation pre-training is an innovative strategy, using text-guided learning to enhance the model's ability to differentiate normality from anomalies.

**Weaknesses:**

The paper lacks clarity on how normality representation pre-training effectively distinguishes between normal and incremental anomalous events.
The continual learning framework's approach to retaining and updating text prompts across anomaly types is not fully explained.
The paper does not thoroughly discuss potential limitations or scalability issues on larger, more complex real-world datasets.

**Questions:**

What specific metrics or comparisons were used to evaluate the mixed-up cross-modal alignment method's success?
Can further explanation be provided on how the framework handles scenarios where anomalies are highly similar to normal events?

---

### Official Review · Reviewer_W7Vj · 2024-11-04

**Soundness:** 3
**Presentation:** 3
**Contribution:** 4
**Rating:** 8
**Confidence:** 4

**Summary:**

This paper explores the field of continual learning for weakly supervised video anomaly detection. The method is designed to cope with scenarios where new anomaly classes are continuously introduced in real applications, and mainly addresses the catastrophic forgetting problem that occurs when existing models learn new anomalies. The method first learns stable normal patterns through pre-training of the normality representation, and subsequently introduces a hybrid cross-modal alignment method that enables the model to perform effective alignment between visual and textual features. Ultimately, the method introduces a learnable textual cueing framework that mitigates the catastrophic forgetting problem by reserving separate textual cues for each abnormality category. Experimental results show that the method outperforms existing continual learning methods on benchmark datasets such as UCF-Crime and XD-Violence.

**Strengths:**

1. This paper proposes a new CL-WSVAD task, which is more valuable for real-world applications.
2. The paper proposes a continual learning method suitable for weakly supervised video anomaly detection, and its effectiveness is reasonably verified through theory and experiments.
3. The paper is well structured and can be easily followed.

**Weaknesses:**

1. In the Normality Representation Pre-training section, 2000 potential anomaly terms were generated using ChatGPT. It would be valuable to examine whether there is any overlap between these terms and the anomalies present in the UCF-Crime and XD-Violence datasets, as this could introduce a degree of information leakage affecting incremental learning.
2. Additionally, it would be insightful to assess how the number of potential anomaly terms generated by ChatGPT impacts the model’s performance and stability.
3. In Task 1, the model undergoes pre-training followed by weakly supervised adaptation. However, it appears that ablation experiments lack a comparison with results obtained without pre-training—i.e., a direct weakly supervised video anomaly detection classification for Task 1, followed by continual learning.
4. It would be beneficial to explore whether continuous learning can be done across datasets, e.g. after learning on the UCF-Crime dataset, continual learning is performed on the XD-Violence dataset and vice versa.

**Questions:**

Discussion of the issues mentioned in the weaknesses is strongly encouraged, as addressing them will make the proposed approach clearer and more effective.

---

### Official Review · Reviewer_Vm5f · 2024-11-05

**Soundness:** 2
**Presentation:** 3
**Contribution:** 2
**Rating:** 5
**Confidence:** 5

**Summary:**

The paper introduces a novel framework for continual learning in weakly supervised video anomaly detection (CL-WSVAD), which emphasizes the streaming anomalies in real-world scenarios. The authors propose a normality representation pre-training mechanism to obtain enhanced representations for normal videos. And a mixed-up cross-modal alignment method is proposed to guide the pre-trained model in achieving effective adaptation on CL-WSVAD. Experimental results on two UCF-Crime and XD-Violence show the effectiveness of the proposed approach.

**Strengths:**

1. This paper proposes a novel framework that combines continual learning with weakly supervised video anomaly detection, addressing the dynamic nature of anomaly streams effectively.

2. The proposed normality representation learning mechanism first obtains potential abnormal texts by ChatGPT, and utilizes pre-trained CLIP to learn the difference between normal videos and potential anomaly categories.

3. The proposed normality coreset memory could fine-tune the normal category in the subsequent task, which enhances the representation of normality in continual learning.

4. The authors design reasonable baselines. The experimental results show the effectiveness of the proposed approach.

**Weaknesses:**

1. Most part of section 3 focuses on video anomaly detection, and the part of anomaly continual learning(section 3.5) should be more detailed.

2. While the framework is designed to mitigate catastrophic forgetting, it seems difficult to demonstrate that catastrophic forgetting has been alleviated from table 1 and table 2.

3. The article does not present any visualization examples to demonstrate the effectiveness of the proposed approach.

4. The paper does not discuss the potential limitations.

**Questions:**

1. How do mixed features work? I am curious about what the mixed features represent semantically? Or is it just a method of data augmentation?

2. How are the “seen testing videos” defined mentioned in section 4.2? Why can the calculated results can highlight the model's ability to mitigate catastrophic forgetting? Perhaps we should evaluate the performance on the test sets of the current task and previous tasks separately?


3. It seems that both Config.1 and Config.2 of continual learning will face a mismatch between the number of normal and abnormal videos during training. Does this phenomenon affect the final result?

---

### Official Review · Reviewer_9rtH · 2024-11-05

**Soundness:** 2
**Presentation:** 3
**Contribution:** 2
**Rating:** 5
**Confidence:** 4

**Summary:**

This paper presents a CLIP-based continual learning framework for the CL-WSVAD task. To prevent catastrophic forgetting, the authors use textual anomaly descriptions for pre-training to enhance normality representation and introduce a mixed-up cross-modal alignment for effective adaptation. Task-specific learnable prompts are retained for each anomaly type to preserve information across tasks. The approach demonstrates comparable performance on a CL-WSVAD benchmark.

**Strengths:**

1. The results on the UCF-Crime dataset are promising, demonstrating the effectiveness of the proposed approach.

2. The CLIP-based framework is intuitive and straightforward, making it easy to understand.

**Weaknesses:**

1. While the authors claim to pioneer the CL-WSVAD paradigm, they appear to overlook prior works such as “Continual Learning for Anomaly Detection in Surveillance Videos” (Doshi and Yilmaz, CVPR 2020 workshop) and “Rethinking Video Anomaly Detection - A Continual Learning Approach” (Doshi and Yilmaz, WACV 2022). A discussion and comparison with these earlier works would strengthen the paper’s novelty claim and situate it more clearly within the context of existing research.

2. The proposed CLIP-based framework shares similarities with existing CLIP-based continual learning approaches, such as Continual-CLIP and AttriCLIP, adapted here for video anomaly detection. Some task-specific design for VAD could enhance the technical contribution and better differentiate this work from existing continual learning frameworks.

**Questions:**

See the weaknesses.

---

### Official Review · Reviewer_DNA7 · 2024-11-09

**Soundness:** 3
**Presentation:** 4
**Contribution:** 3
**Rating:** 6
**Confidence:** 5

**Summary:**

This paper presents a new approach to weakly supervised video anomaly detection in a streaming context, where new types of anomalies are introduced continually. The authors argue that existing Weakly Supervised Video Anomaly Detection (WSVAD) methods, which train on a fixed set of anomalies, struggle to generalize to real-world scenarios where anomalies appear incrementally. To address this, they propose a framework that adapts CLIP-based vision-language models to continually learn new anomaly types while retaining knowledge of previously encountered anomalies, thereby mitigating catastrophic forgetting. Hence, if the authors revise their paper, I am pleased to give a positive judgment:
1、The paper assumes that each task introduces only one anomaly type at a time. However, in real-world settings, the types of anomalies could increase at a faster rate. The scalability of the method to handle such rapid changes remains untested.
2、The approach is validated primarily on two specific datasets (UCF-Crime and XD-Violence), which may limit generalizability to other datasets with different types of anomalies and environmental contexts. Testing the method on more diverse datasets could further substantiate its broad applicability.
3、The pre-training stage uses negative anomaly descriptions generated by ChatGPT, but the realism and quality of these synthetic texts could impact the generalization ability of normality representations, especially if the anomaly descriptions lack detail or specificity.
4、The method introduces several loss functions (e.g., cross-modal alignment loss, textual semantic contrastive loss), which come with hyperparameters that require careful tuning. Further analysis of the contribution of each loss component to the final performance would help clarify the role of each module.
5、Although the method pre-trains on normal representations, it may still struggle to distinguish new normal events from potential anomalies, especially in diverse real-world scenarios. The ability of the model to handle ambiguous cases could benefit from further exploration.
6、The experimental results rely on several hyperparameters, which may vary in effectiveness across different distributions or environments. Testing the method’s robustness with varied settings could provide more insights into its adaptability and stability in other contexts.

**Strengths:**

This paper presents a new approach to weakly supervised video anomaly detection in a streaming context, where new types of anomalies are introduced continually. The authors argue that existing Weakly Supervised Video Anomaly Detection (WSVAD) methods, which train on a fixed set of anomalies, struggle to generalize to real-world scenarios where anomalies appear incrementally. To address this, they propose a framework that adapts CLIP-based vision-language models to continually learn new anomaly types while retaining knowledge of previously encountered anomalies, thereby mitigating catastrophic forgetting.

**Weaknesses:**

1、The paper assumes that each task introduces only one anomaly type at a time. However, in real-world settings, the types of anomalies could increase at a faster rate. The scalability of the method to handle such rapid changes remains untested.
2、The approach is validated primarily on two specific datasets (UCF-Crime and XD-Violence), which may limit generalizability to other datasets with different types of anomalies and environmental contexts. Testing the method on more diverse datasets could further substantiate its broad applicability.
3、The pre-training stage uses negative anomaly descriptions generated by ChatGPT, but the realism and quality of these synthetic texts could impact the generalization ability of normality representations, especially if the anomaly descriptions lack detail or specificity.
4、The method introduces several loss functions (e.g., cross-modal alignment loss, textual semantic contrastive loss), which come with hyperparameters that require careful tuning. Further analysis of the contribution of each loss component to the final performance would help clarify the role of each module.
5、Although the method pre-trains on normal representations, it may still struggle to distinguish new normal events from potential anomalies, especially in diverse real-world scenarios. The ability of the model to handle ambiguous cases could benefit from further exploration.
6、The experimental results rely on several hyperparameters, which may vary in effectiveness across different distributions or environments. Testing the method’s robustness with varied settings could provide more insights into its adaptability and stability in other contexts.

**Questions:**

1、The paper assumes that each task introduces only one anomaly type at a time. However, in real-world settings, the types of anomalies could increase at a faster rate. The scalability of the method to handle such rapid changes remains untested.
2、The approach is validated primarily on two specific datasets (UCF-Crime and XD-Violence), which may limit generalizability to other datasets with different types of anomalies and environmental contexts. Testing the method on more diverse datasets could further substantiate its broad applicability.
3、The pre-training stage uses negative anomaly descriptions generated by ChatGPT, but the realism and quality of these synthetic texts could impact the generalization ability of normality representations, especially if the anomaly descriptions lack detail or specificity.
4、The method introduces several loss functions (e.g., cross-modal alignment loss, textual semantic contrastive loss), which come with hyperparameters that require careful tuning. Further analysis of the contribution of each loss component to the final performance would help clarify the role of each module.
5、Although the method pre-trains on normal representations, it may still struggle to distinguish new normal events from potential anomalies, especially in diverse real-world scenarios. The ability of the model to handle ambiguous cases could benefit from further exploration.
6、The experimental results rely on several hyperparameters, which may vary in effectiveness across different distributions or environments. Testing the method’s robustness with varied settings could provide more insights into its adaptability and stability in other contexts.

---

### Author Response · Authors · 2024-11-28
**General Response**

We sincerely appreciate the detailed and constructive feedback provided by all the reviewers.

Specifically, we are delighted that the reviewers have acknowledged our substantial contributions (W7Vj, DNA7, zzoQ, JakS). We are also pleased that the reviewers recognized the novelty of our work (W7Vj, DNA7, zzoQ, Vm5f) and the effectiveness of our method (W7Vj, 9rtH, Vm5f, oFsJ), which is supported by a solid motivation (oFsJ) and a clear presentation (DNA7).
Here, we briefly summarize the manuscript changes and recurring points highlighted in the feedback:

1. Enhanced Related Work Discussion:

- The advantages of our method over the WSVAD and UVAD paradigms. The comparison with existing continual learning approaches for UVAD.
- The task-specific designs for VAD in our method compared to other CLIP-based approaches.

2. Quantitative Evaluation:

- Continual learning experiments with multi-class anomaly setting.
- Analysis of the number of anomaly texts in NRP.
- Analysis of the content of anomalous texts in NRP.
- Performance validation on cross-dataset experiments.
- Visualization experiments for verifying the effectiveness.

3. Extended Evaluation:

- The ablation studies on NRP and MCMA have been expanded.
- Performance validation in additional metrics.
- Analysis of the hyperparameters of the loss function.

4. More Comprehensive Explanation:

- The principles of our method for distinguishing new normal events from potential anomalies or handling scenarios where anomalies are highly similar to normal events.
- The principle of NRP, MCMA, and the method of updating learnable text prompts.
- The more solid explanation of our setting.
- The impact of the imbalance between normal and abnormal videos on the final results.
- The limitations of our method.

5. Others:

- The proof process of NRP has been improved.
- Section 3.5 have been more detailed. The explanation regarding retaining and updating learnable text prompts has been enriched.
- Typos have been corrected.

---

### Author Response · Authors · 2024-12-01
**General Response**

We would like to express our gratitude to all the reviewers for their valuable feedback. We have carefully considered all suggestions and updated our submission accordingly. With only two days remaining for discussion, we kindly request the reviewers' feedback. We believe that constructive and timely communication between reviewers and authors is essential for the benefit of both parties.

---

### Author Response · Authors · 2024-12-02
**General Response**

Dear Reviewers,

We sincerely appreciate the time and effort you have dedicated to reviewing our manuscript and offering valuable feedback.  We have responded to the comments and updated our submission accordingly.
As the author-reviewer discussion phase is drawing to a close, we would like to confirm whether our responses have effectively addressed your concerns. We are more than willing to continue our communication with you.

Best regards,

The authors of Paper 3776

---

### Meta-Review · Area_Chair_hrnp · 2024-12-18

**Metareview:**

This paper proposed a framework of continual learning for weakly supervised video anomaly detection (CL-WSVAD), assuming that all normal videos are collected during the initial stage and new types of anomalies are introduced continually. To overcome catastrophic forgetting, abnormal type specific prompt is learned. The strength is that the proposed method is well motivated and straightforward, achieving promising performance on UCF-Crime. The weakness is that the setting of the proposed CL-WSVAD is unrealistic and the evaluation is limited, hindering its generalization in real-world applications.

**Additional Comments On Reviewer Discussion:**

The concerns raised by reviewers focus on the unrealistic task setting, limited evaluation, lack of sensitivity analysis of hyperparameters and comparison with prior related works, etc. During rebuttal, the authors provided more related work discussion, more quantitative evaluation, and comprehensive hyperparameter sensitivity analysis. However, the setting that each new type of anomalies is introduced continually is unrealistic. Although experimental results on the other settings are provided, it is unclear how the Normality Representation updated if new normal video is also introduced continually. Therefore, I recommend another round of revision to incorporate more realistic scenarios provided by the insightful reviewers.

---

### Decision · Program_Chairs · 2025-01-22

Reject